Genome-wide investigation and expression pattern of PHR family genes in cotton under low phosphorus stress

Zhao Yan
Li Peiyu
Wang Huarui
Feng Jiping
Li Yuxin
Wang Shanshan
Li Yuanjie
Guo Yanyan
Li Lin
Su Yao 254211823@qq.com
Sun Zhengwen nxszhw@hebau.edu.cn
State Key Laboratory of North China Crop Improvement and Regulation, Key Laboratory for Crop Germplasm Resources of Hebei, College of Agronomy, Hebei Agricultrual University , Baoding, China , China
Cao Yunpeng
Electronic publication date: 2022 Dec 15
Publication date: 2022
Volume: 10
Electronic Location ID: e14584
Received 2022 Jun 14; Accepted 2022 Nov 28
Copyright: © 2022 Zhao et al.
Copyright year: 2022
Copyright holder: Zhao et al.
License: This is an open access article distributed under the terms of the Creative Commons Attribution License, which permits unrestricted use, distribution, reproduction and adaptation in any medium and for any purpose provided that it is properly attributed. For attribution, the original author(s), title, publication source (PeerJ) and either DOI or URL of the article must be cited.
License URL: https://creativecommons.org/licenses/by/4.0/

Keywords: Cotton, PHR gene, Transcription factor, Low phosphorus stress, Expression

Funding: The work was supported by Key Scientific and Technological Research Projects of University in Hebei Province (QN2021073), the Key Research and Development Program of Hebei Province (21326314D), the Startup Fund from Hebei Agricultural University (YJ201824), and the Training Program of Innovation and Entrepreneurship for Undergraduates of Hebei Agricultural University (2020270). The funders had no role in study design, data collection and analysis, decision to publish, or preparation of the manuscript The work was supported by Key Scientific and Technological Research Projects of University in Hebei Province (QN2021073), the Key Research and Development Program of Hebei Province (21326314D), the Startup Fund from Hebei Agricultural University (YJ201824), and the Training Program of Innovation and Entrepreneurship for Undergraduates of Hebei Agricultural University (2020270). The funders had no role in study design, data collection and analysis, decision to publish, or preparation of the manuscript.

==============================
Phosphorus starvation response (PHR) protein is an important transcription factor in phosphorus regulatory network, which plays a vital role in regulating the effective utilization of phosphorus. So far, the PHR genes have not been systematically investigated in cotton. In the present study, we have identified 22, 23, 41 and 42 PHR genes in G. arboreum, G. raimondii, G. hirsutum and G. barbadense, respectively. Phylogenetic analysis showed that cotton PHR genes were classified into five distinct subfamilies. The gene structure, protein motifs and gene expression were further investigated. The PHR genes of G. hirsutum from the same subfamily had similar gene structures, all containing Myb_DNA-binding and Myb_CC_LHEQLE conserved domain. The structures of paralogous genes were considerably conserved in exons number and introns length. The cis-element prediction in their promoters showed that genes were not only regulated by light induction, but also were related to auxin, MeJA, abscisic acid-responsive elements, of which might be regulated by miRNA. The expression analysis showed that the GhPHR genes were differentially expressed in different tissues under various stresses. Furthermore, GhPHR6, GhPHR11, GhPHR18 and GhPHR38 were significantly changed under low phosphorus stress. The results of this study provide a basis for further cloning and functional verification of genes related to regulatory network of low phosphorus tolerance in cotton.

Introduction

Phosphorus (P) is a necessary mineral nutrient for plant growth and development, playing an important role in plant cell energy metabolism, enzyme reaction and signal transduction. Plants mainly absorb inorganic phosphorus from soil solution through roots. However, about 95~99% of phosphorus easily complexes with iron, calcium and aluminum in acidic and alkaline soils to produce insoluble phosphorus, which is not conducive to plant absorption and assimilation, becoming primary constraint to global crop production (Péret et al., 2011). For crop production, it is necessary to apply excessive phosphorus fertilizer to supplement the available phosphorus in the soil, but this process will cause serious environmental pollution. Therefore, for sustainable production in phosphorus deficient soil, it is necessary to explore and analyze the molecular basis of high-efficiency utilization of available phosphorus.

To adapt to a low phosphorus medium, plants have evolved a set of complex gene regulation networks, among which the most important members related to phosphate absorption and transport involved PHR1 (phosphate startup response 1), IPS1 (induced by phosphate startup 1), miR399 (microRNA399), PHO2 (phosphate 2) and PT (phosphate transport). In this complex regulatory network, PHR protein is an important transcription factor in plant phosphorus regulatory network, which plays a key role in signal transduction and regulation induced by phosphate starvation. At present, PHR genes of Arabidopsis thaliana, Oryza sativa, Zea mays, Glycine max and other species have been identified (Bustos et al., 2010; Woo et al., 2012; Lin et al., 2013; Guo et al., 2015; Xue et al., 2017; Xu et al., 2018). Arabidopsis PHR transcription factor directly regulates gene expression by binding to the sequence of phosphorus starvation induced gene P1BS (GNATATNC), and there is functional redundancy among members (Rubio et al., 2001). Arabidopsis PHR1 and PHL1 (PHR1-LIKE) transcription factors play a role in plant response to phosphorus starvation. PHR1 can bind to the promoters of many phosphorus starvation induced genes. The loss of AtPHR1 gene function in Arabidopsis will reshape membrane lipid metabolism, primary and secondary metabolism and photosynthesis, which will affect the growth rate of Arabidopsis root and crown and the accumulation of anthocyanins. PHR1 deletion mutation will affect the expression of some phosphorus starvation induced genes, resulting in the decrease of glucose, fructose, sucrose and starch contents in the mutants (Bustos et al., 2010; Nilsson et al., 2012; Pant et al., 2015). In rice, SPX family proteins are involved in phosphorus sensing and signaling by inhibiting the transcriptional activity of OsPHR2 (Lv et al., 2014). Rice contains genes OsPHR1, OsPHR2 and OsPHR3 with MYB-CC domain. The loss of any gene function will inhibit the elongation of rice root hair, and then affect the effective absorption of phosphate by plants. MiRNAs regulate plant response to low phosphorus by down regulating gene transcription (Zeng et al., 2014). MiR399 and miR827 are involved in the response of plants to low phosphorus stress (Pant et al., 2008; Lin et al., 2010). MiR399-PHO2 and miR827-NLA mediate the ubiquitination and degradation of phosphate transporters PHT1 and PHO1, and participate in the systematic regulation of phosphorus balance (Chiou et al., 2006; Liu et al., 2014).

Cotton is an important cash crop and raw material for textile industry in China, and plays an important role in the national economy (Ma et al., 2021). Phosphorus is one of the three necessary nutrient elements for cotton growth and development. It can promote cotton budding and flowering in the middle growth stage, promote cottonseed maturation, increase boll weight and open boll early in the later growth stage, which directly affects the yield and fiber quality of seed cotton. Under low phosphorus stress, the adaptive change of root morphology is an important biological basis for crops to make efficient use of soil phosphorus. It has been found the total root length, total root surface area, lateral root length and lateral root number increase in varying degrees under low phosphorus stress in wheat, maize and rice (Chiou & Lin, 2011; Guo et al., 2015; Sawers et al., 2017). When low phosphorus stress occurs, different crops will form a set of adaptive mechanisms to deal with stress. However, there are few studies on the response to low phosphorus stress of cotton (Lei et al., 2022).

In order to explore the candidate genes of low phosphorus tolerance in cotton, we identified the members of PHR gene family, and analyzed their gene structure, cis-acting elements and expression pattern based on four Gossypium genomes including the latest published genomes of allotetraploid cotton. It provides a reference for further revealing the biological function of PHR transcription factor under low phosphorus stress in cotton.

Material and method

Identification and analysis of cotton PHR gene family members

To obtain the members of PHR family genes, the amino acid sequences with conserved domain Myb DNA-binding (PF00249) and Myb_CC_LHEQLE (PF14379) of the PHR transcription factor family were searched from the four Gossypium genomes including Gossypium arboreum (CRI, version 1.0), G. raimondii (JGI, version 2.0), G. hirsutum(HEBAU, version 1.0) and G. barbadense (HEBAU, version 1.0) (Paterson et al., 2012; Wang et al., 2012; Li et al., 2014; Ma et al., 2021). The obtained sequences of PHR transcription factor were identified using SMART (http://smart.embl-heidelberg.de/) and CDD databases (https://www.ncbi.nlm.nih.gov/Structure/cdd/wrpsb.cgi). The physical and chemical properties of the proteins were predicted using the online website ExPASY (https://web.expasy.org/protparam/).

Conserved domain and gene structure analysis of cotton PHR family members

The online website MEME (http://meme-suite.org/tools/meme) was used to predict the conserved motif of cotton PHR genes, and the number of motifs was set to 10, other parameters are the default settings. The genes structure of cotton PHR transcription factors were draw using online website GSDS (http://gsds.gao-lab.org/) based on cotton genome annotations.

Phylogenetic relationship of cotton PHR gene family members

The protein sequences of Arabidopsis thaliana PHR family gene were downloaded from Phytozome database (https://phytozome-next.jgi.doe.gov). The protein sequences of PHR family genes in cotton and Arabidopsis were compared by MEGA 11.0 software (Koichiro, Glen & Sudhir, 2021), and the phylogenetic tree was constructed by adjacency method (NJ). The bootstrap value was 1,000, the model is Poisson model. Finally, we showed the results using the online tool iTOL (Letunic & Bork, 2021).

Analysis of cis-acting elements of promoters of GhPHR gene family members

To understand the possible regulation and response mechanism of cotton PHR genes, the 1.5 kb upstream of genome sequences were obtained from each GhPHRs, and submitted to PlantCARE website (http://bioinformatics.psb.ugent.be/webtools/plantcare/html/) to predict cis-acting elements of promoters, and finally visualized by TBtools software (Chen et al., 2020).

Prediction of miRNA-target GhPHR genes

According to the principle of sequence complementarity, the regulatory miRNAs of GhPHR genes were predicted. The GhPHR target genes of miRNA were predicted using psRNATarget online software (https://www.zhaolab.org/psRNATarget/).

Differential expression analysis of GhPHR family members in roots under low phosphorus stress

Tissue expression specificity of GhPHR genes were analyzed by using RNA sequencing data of G. hirsutum ‘TM-1’ (Zhang et al., 2015) during different development stages and stress treatments downloaded from the NCBI Sequence Read Archive (PRJNA490626). To further screen GhPHR genes in response to low phosphorus stress, we generated and analyzed the genes expression in roots based on transcriptome data of a P-resistant accession under P deficient hydroponic conditions. The cotton seeds were grown in germination boxes containing quartz sand, and the seedlings were moved into half-strength Hoagland normal nutrient solution when the cotyledons had fully expanded. After 3 days, the nutrient solution was replaced with P-deficient and P-replete Hoagland nutrient solution, respectively. The roots of three cotton seedings were sampled at 0 day and 15 days for RNA-seq. The gene expression patten was drawn by Hemi 1.0 software (Deng et al., 2014) with log2 (1 + FPKM) values after averaging three replicates. The expression of six GhPHR genes were selected for further verification by qRT-PCR. All reactions were performed using the Roche LightCycler96 RealTime PCR System with three independent biological replicates, each with three technical replicates. The expression of GhUBQ14 was used as the internal control. Relative gene expression values were calculated using the 2−ΔCT method (Schmittgen & Livak, 2008).

Results

Identification of cotton PHR family gene members

To investigate the copy number variation of the PHR genes during cotton evolution, a comprehensive search was conducted for PHR genes across four cottons including G. arboreum, G. raimondii, G. hirsutum and G. barbadense. A total of 128 PHR gene sequences were detected in the four cotton species, 22, 23, 41, and 42 PHR genes were identified, respectively. The detailed information was listed in Tables S1 and S2. The results were further verified through the NCBI-CDD and SMART database. The results showed that the numbers of PHR genes in two diploid cotton species were almost similar as were those in two tetraploid cotton species. The number of PHR family genes in G. arboretum and G. raimondii were basically half in G. hirsutum and G. barbadense, indicating that the PHR family was relatively conserved, which conforms to the known evolutionary relationship in different cottons (Paterson et al., 2012).

The names of the PHR genes were determined according to the gene information of Arabidopsis and the locations on the chromosomes. The encoded protein of the PHR family genes in G. hirsutum contains 236~494 amino acid residues. The relative molecular mass is between 26.53 and 54.30 kDa, and the theoretical isoelectric point is between 5.50 and 9.82. Each of the family members contains Myb DNA-binding and Myb_CC_LHEQLE domain. Fourty-one PHR genes of G. hirsutum were distributed on 21 chromosomes except A05, A07, D01, D03 and D07 (Table 1). Subgenome A and subgenome D contained 22 and 19 sequences, respectively. The genes number in subgenome A was consistent with the genes number of GaPHR, and four GhPHR genes from subgenome D were missing compared to GrPHRs. This result indicated that subgenome D of G. hirsutum might have lost genes due to redundant gene functions during evolution.

Table 1 Information of PHR gene family in G. hirsutum.

A subgenome of G. hirsutum	D subgenome of G. hirsutum	
Gene name	Gene ID	Protein length	MV (Da)	pI	Gene name	Gene ID	Protein length	MV (Da)	pI	
GhPHR1	GhM_A01G2399	380	42,304.2	7.76	GhPHR23	GhM_D02G1490	364	40,435.4	6.85	
GhPHR2	GhM_A02G0277	303	33,075.2	6.52	GhPHR24	GhM_D04G2388	346	38,667.5	9.37	
GhPHR3	GhM_A03G1365	364	40,578.5	6.85	GhPHR25	GhM_D05G1302	494	54,301.4	6.23	
GhPHR4	GhM_A04G1918	346	38,712.5	9.34	GhPHR26	GhM_D06G2662	298	32,958.0	5.88	
GhPHR5	GhM_A06G2674	298	32,865.9	6.11	GhPHR27	GhM_D06G2663	333	35,805.0	8.65	
GhPHR6	GhM_A06G2675	333	35,829.1	8.65	GhPHR28	GhM_D08G0228	411	46,409.8	7.05	
GhPHR7	GhM_A08G0240	417	47,059.5	6.86	GhPHR29	GhM_D08G1976	279	31,618.2	9.82	
GhPHR8	GhM_A08G2024	279	31,659.3	9.78	GhPHR30	GhM_D08G2683	372	41,765.9	8.19	
GhPHR9	GhM_A08G2740	411	46,419.1	8.69	GhPHR31	GhM_D08G2924	357	39,650.8	8.62	
GhPHR10	GhM_A08G2986	348	38,638.6	8.77	GhPHR32	GhM_D09G1508	316	36,391.6	8.20	
GhPHR11	GhM_A09G1611	315	36,353.5	8.05	GhPHR33	GhM_D09G1943	267	30,049.4	8.20	
GhPHR12	GhM_A09G2040	267	29,998.3	8.20	GhPHR34	GhM_D10G0017	399	44,484.4	5.78	
GhPHR13	GhM_A09G2485	253	27,983.4	5.97	GhPHR35	GhM_D10G1602	302	33,108.2	6.78	
GhPHR14	GhM_A10G0026	433	48,090.4	5.50	GhPHR36	GhM_D11G1558	478	52,511.1	5.69	
GhPHR15	GhM_A10G1517	302	33,070.1	6.25	GhPHR37	GhM_D11G3020	448	49,301.5	6.20	
GhPHR16	GhM_A11G1564	478	52,622.2	5.69	GhPHR38	GhM_D11G3158	387	43,073.3	7.41	
GhPHR17	GhM_A11G3088	418	46,179.0	5.97	GhPHR39	GhM_D12G2463	236	26,531.7	8.43	
GhPHR18	GhM_A11G3236	387	43,196.4	6.88	GhPHR40	GhM_D13G0849	356	39,260.4	8.15	
GhPHR19	GhM_A12G2578	236	26,730.9	8.43	GhPHR41	GhM_D13G1605	347	38,580.7	8.01	
GhPHR20	GhM_A13G0904	309	34,021.4	8.07						
GhPHR21	GhM_A13G1449	374	41,288.3	8.17						
GhPHR22	GhM_A13G1727	347	38,480.6	8.33						

Three GhPHR genes were observed on chromosomes A09 and A13, while D09 and D13 only contained two genes. The A01 and A03 chromosomes contained one gene, but no GhPHR gene sequence was contained in D01 and D03. These results showed that the GhPHR genes might have been lost and duplicated in the process of cotton evolution, indicating a strong correlation between subgenome A and subgenome D (Paterson et al., 2012; Wang et al., 2018).

Phylogenetic analysis of the PHR gene family in cotton

To explore the phylogenetic relationship of the cotton PHR genes, we constructed a phylogenetic tree with the neighbor-joining method using 41 G. hirsutum, 42 G. babardence, 22 G. arboreum, 23 G. raimondii and 13 Arabidopsis PHR amino acid sequences (Fig. 1). All the PHR genes can be divided into five subgroups. The PHR genes number of G. hirsutum and G. barbadense was basically twice of G. arboreum and G. raimondii in each subgroup. This result was consistent with the previous analysis and conformed to the conserved evolutionary relationship in cottons. Among these subgroups, the largest subgroup V consisted of 12 GhPHRs, 12 GbPHRs, seven GaPHRs, seven GrPHRs and six AtPHRs, showing that has expanded considerably in two allotetraploid cottons. In contrast, subgroup III only included four GhPHRs, four GbPHRs, two GaPHRs, two GrPHRs and one AtPHRs, indicating might be a highly-conserved clade.

Figure 1 Phylogenic tree of the PHR family members in G. arboreum, G. raimondii, G. hirsutum, G. barbadense and Arabidopsis thaliana.

The unrooted phylogenic tree was constructed using MEGA 11.0 by neighbor-joining method. Numbers on branches were bootstrap portions from 1,000 replicates. The subgroups were marked in different colors.

Analyses of gene structures and protein motifs of PHR genes in G. hirsutum

Gene structure analysis of PHR gene family members showed that the gene structure of GhPHR members in the same subgroup was similar, and there was little difference in the number of exons of GhPHR members among different subgroups. Except for the GhPHR genes of class V which contained seven exons, most of the other GhPHR genes contained six exons (Fig. 2). Furthermore, these PHR protein sequences were submitted to MEME to discover conserved motifs. The adjacent clades carried similar motifs. Analysis of the conserved domains of GhPHR gene family members showed that Myb_DNA-binding and Myb_CC_LHEQLE were present in all GhPHR proteins and other motifs were functionally unknown motifs (Fig. 2). Among these, motif 4 occurred in class I, class II and class IV subgroups, motif 6 was unique to class II subgroup, motif 10 were unique to class IV subgroup, and motif 5 was unique to two of class I class II subgroup. It is worth noting that the GhPHR protein motif types of class II were different. Among them, there were eight motifs in GhPHR2, GhPHR15 and GhPHR35, only six motifs in GhPHR5 and GhPHR26, and seven motifs in the other five GhPHRs. These special conserved motifs may be the main factors enabling GhPHRs to participate in different biological functions.

Figure 2 Distributions of gene structure and conserved protein motifs in GhPHR genes.

(A) Gene structure of all GhPHR genes. (B) Conserved protein motifs of all GhPHR genes. The red boxes and gray lines represent the exon and intron, respectively. The lengths of the boxes and lines were scaled based on the length of the genes. Conserved motifs in the GhPHR proteins are indicated by colored boxes.

Analysis of cis-acting elements in the promoter of GhPHR family genes

To further clarify the possible regulatory mechanism of GhPHR family genes under abiotic stress, the promoter sequences were analyzed by using PlantCARE database. The results showed 13 types of cis-acting elements, including light responsiveness, salicylic acid responsiveness, gibberellin-responsiveness, MeJA-responsiveness, anaerobic induction, auxin-responsiveness, abscisic acid responsiveness and so on (Fig. 3). In terms of composition and quantity, the GhPHR genes contained an average of 18 cis-acting elements, all of which had light responsiveness elements. Among them, GhPHR6 and GhPHR19 contained the most types of response elements (10 kinds), while GhPHR3, GhPHR5, GhPHR7 and GhPHR40 contained the least cis-acting elements (three kinds). In terms of element types, 17 genes contained gibberellin—responsiveness elements including GhPHR1, GhPHR12 and GhPHR34. Thirty-one genes contained anaerobic induction elements including GhPHR5, GhPHR19, GhPHR24, and 11 genes contained auxin-responsiveness elements including GhPHR16, GhPHR20, GhPHR36. The results showed that the GhPHR genes were not only regulated by light induction, but also played a role during drought, anaerobic and other stresses. Among them, the promoter regions of GhPHR3, GhPHR5, GhPHR7 and GhPHR40 contained less cis elements, which were only related to light, MeJA, abscisic acid and anaerobic induction.

Figure 3 Cis-acting element analysis of PHR family members in G. hirsutum.

The colored boxes indicate different cis-elements in promoters of genes.

Prediction of the regulatory miRNA of PHR gene family

The online software psRNATarget was used to predict and analyze the regulatory miRNAs of GhPHR genes. The regulatory combinations of 33 miRNAs and PHR genes were predicted (Table S3). It was found that 12 miRNAs could regulate 16 PHR genes. Unpaired energy (UPE) is the energy required to unlock the secondary structure of the target gene miRNA target site. A lower UPE value indicates that miRNA is more likely to bind or cleave the target gene. This study showed some of the predicted results with an expected value less than or equal to 5. GhPHR21 and GhPHR32 can be recognized by miR396 and miR7510a, miR2949 and miR7491 at the same time, respectively, which may be regulated by these two miRNAs. GhPHR19 may be regulated by the sequence cleavage of miR482, GhPHR4 and GhPHR24 may be regulated by the transcriptional inhibition of miR827, GhPHR17 and GhPHR37 may be regulated by the sequence cleavage of miR2948-5p, and GhPHR23, GhPHR25 and GhPHR40 may be regulated by the transcriptional inhibition of miR2948-5p. In this study, the regulation modes of interaction combinations were different. Approximately 2/3 of the regulation modes belonged to sequence cleavage and 1/3 belonged to transcriptional inhibition (Table 2).

Table 2 Bioinformatic analysis of partial miRNA target sites.

miRNA	Target genes	Expectation		Start	Sequence	End	Inhibition mode	
miR827a	GhPHR4	5.0	miRNA	1	UUAGAUGACCAUCAACAAACA	21	Translation	
					: ::::::: ::::::.:.:			
			Target	952	GGAUUGUUGA-GGUCAUUUGA	971		
miR827b	GhPHR4	5.0	miRNA	1	UUAGAUGACCAUCAACAAACA	21	Translation	
					: ::::::: ::::::.:.:			
			Target	952	GGAUUGUUGA-GGUCAUUUGA	971		
miR827c	GhPHR4	5.0	miRNA	1	UUAGAUGACCAUCAACAAACA	21	Translation	
					: ::::::: ::::::.:.:			
			Target	952	GGAUUGUUGA-GGUCAUUUGA	971		
miR7491	GhPHR11	4.0	miRNA	1	UGGGAUCUUCGAGAGGAUUGAGCC	24	Translation	
					::::::.: :::::::.:			
			Target	324	CCAGAAAUCCUUUGAAAGAUCCUA	347		
miR2949b	GhPHR12	4.0	miRNA	1	UCUUUUGAACUGGAUUUGCCGA	22	Translation	
					..:.:::: :::::::::			
			Target	497	AGUCUGAGUCCAAUUCAAAAGA	518		
miR2949c	GhPHR12	4.0	miRNA	1	UCUUUUGAACUGGAUUUGCCGA	22	Translation	
					..:.:::: :::::::::			
			Target	497	AGUCUGAGUCCAAUUCAAAAGA	518		
miR2949a-5p	GhPHR12	5.0	miRNA	1	ACUUUUGAACUGGAUUUGCCGA	22	Translation	
					..:.:::: ::::::::			
			Target	497	AGUCUGAGUCCAAUUCAAAAGA	518		
miR482a	GhPHR19	5.0	miRNA	1	UCUUUCCUACUCCUCCCAUACC	22	Cleavage	
					::::: .:::: :::.:::			
			Target	620	AUUAUGGAGGGAGAUGGAGAGA	641		
miR7510a	GhPHR21	4.5	miRNA	1	AAGGUCAUGAUCUUUAGCGGCGUU	24	Cleavage	
					::.: : ::..::::. ::.::::			
			Target	88	AAUGGCACUGGAGAUUCUGGCCUU	111		
miR396a	GhPHR21	5.0	miRNA	1	UUCCACAGCUUUCUUGAACUG	21	Cleavage	
					:: ::::::.::. :::::			
			Target	560	AAGCUCAAGAGAGUCUUGGAA	580		
miR396b	GhPHR21	5.0	miRNA	1	UUCCACAGCUUUCUUGAACUG	21	Cleavage	
					:: ::::::.::. :::::			
			Target	560	AAGCUCAAGAGAGUCUUGGAA	580		
miR2948-5p	GhPHR40	4.5	miRNA	1	UGUGGGAGAGUUGGGCAAGAAU	22	Translation	
					::: ::::.. :::::..:::			
			Target	46	UUUCCUGCCUGCCUCUCUUACA	67		

Tissue expression analysis of GhPHR family genes

The expression of GhPHR family genes was investigated across different tissues and developmental stages of upland cotton. Most of these genes were expressed at varying levels across different tissues and developmental stages (Fig. 4). It was found that five GhPHR genes were the most commonly expressed in all tissues. A total of 13 GhPHRs were highly expressed in all tissues, indicating that these genes play an important role in all morphogenesis of cotton. Among them, GhPHR5 and GhPHR6 were highly expressed in leaves, while GhPHR13, GhPHR14, GhPHR26 and GhPHR27 were highly expressed in both roots and leaves. Combined with the cis-element structure of GhPHR promoters, it was speculated that they may play an important role in leaf photosynthesis. A total of eight GhPHRs were expressed across different tissues except fiber developmental stages, and the expression of other six GhPHR genes were low in different tissues. Altogether, the expression profiles of GhPHR genes showed that it plays a role in different tissues of cotton, among which GhPHR2, GhPHR5, GhPHR6 and GhPHR13 have obvious tissue expression specificity.

Figure 4 Expression profiles of GhPHR genes in different tissues.

The color represents the gene expression level based on public transcriptome data. The red and blue colors indicate a high and a low expression level, respectively.

Expression pattern of GhPHR genes in roots under low phosphorus stress

The expression of 41 GhPHR genes in roots showed that most GhPHR genes were affected under low phosphorus treatment, except that GhPHR4, GhPHR7, GhPHR12, GhPHR19, GhPHR24, GhPHR33 and GhPHR39 were not detected (Fig. 5). The expression of GhPHR1 and GhPHR11 decreased under low phosphorus stress but that of GhPHR3, GhPHR6, GhPHR17, GhPHR18, GhPHR27, GhPHR30 and GhPHR38 increased. Among them, expression level of GhPHR17, GhPHR30 and GhPHR38 was significantly higher than that before stress treatment. In addition, GhPHR5, GhPHR13, GhPHR15 and GhPHR26 maintained a high expression level. It should be noted that the expression of GhPHR11 was significantly lower under low phosphorus stress than under normal phosphorus treatment, and that of GhPHR18 was significantly higher than that of normal phosphorus treatment (Fig. 5). Further, we selected six differentially expressed GhPHR genes for verification by qRT-PCR, showing a consistent trend (Fig. 6). This provided further evidence that the six putative genes were closely associated with low-phosphorus tolerance.

Figure 5 Expression pattern of GhPHR genes in root under low phosphorus stress.

(A) Expression level of PHR genes at 0 d and 15 d under low and normal phosphorus condition, respectively. (B) Cotton seedlings of 15 d under low and normal phosphorus condition. The colorful scale of heat map indicates the relative expression levels where blue indicates low and red indicates high.

Figure 6 Relative expression level of six representative GhPHR genes.

(A) GhM_A01G2399. (B) GhM_A06G2675. (C) GhM_A09G1611. (D) GhM_A11G3236. (E) GhM_D08G2683. (F) GhM_D11G3158. Data represent mean ± SE of three biological replicates by qRT-PCR.

Discussion

Phosphorus deficiency is a major factor limiting crop yield. In cotton, the PHR genes have not been systematically investigated. In the present study, we have identified 22, 23, 41 and 42 PHR genes in G. arboreum, G. raimondii, G. hirsutum and G. barbadense, respectively. The GhPHR genes are differentially expressed in different tissues under various stresses. Furthermore, GhPHR6, GhPHR11, GhPHR18 and GhPHR38 were significantly changed under low phosphorus stress. These results provided a basis for low phosphorus tolerance in cotton.

Plants have evolved a series of morphological, physiological and molecular strategies to adapt to phosphorus deficiency (Veneklaas et al., 2012), including symbiosis with mycorrhizal fungi, secretion of organic acids, remodeling of root structure, and improving the expression of phosphorus transporters (Chiou & Lin, 2011; Sawers et al., 2017). Most of these strategies improve the utilization efficiency of phosphorus by enhancing the mobility of phosphorus in soil or the acquisition of phosphorus by roots. In recent years, genes and proteins related to low phosphorus stress have been found and identified. Among them, PHR is a MYB transcription factor, which plays an important role in plant response to low phosphorus stress (Bustos et al., 2010; Sega & Pacak, 2019). It has been reported that PHR1 and PHR1-like genes play a key role in the phosphorus signal regulation network of plants such as Arabidopsis (Karthikeyan et al., 2007), rice (Guo et al., 2015), soybeans (Xue et al., 2017), wheat (Chiou & Lin, 2011), maize (Lin et al., 2013; Sawers et al., 2017) and rape (Ren et al., 2012). In addition, genome-wide transcriptional analysis of Arabidopsis and rice showed that most phosphorus starvation response genes were induced and activated by AtPHR1 and OsPHR2 and their homologous genes AtPHL1, AtPHL2, OsPHR1 and OsPHR3 (Guo et al., 2015; Sun et al., 2016). In our study, the expression of GhPHR1 and GhPHR11 decreased but that of GhPHR6, GhPHR18, GhPHR30 and GhPHR38 increased under low phosphorus stress, we further verified the six genes by qRT-PCR, indicating closely associated with low-phosphorus tolerance.

Cis-acting elements regulate gene transcription by responding to different external signals, and then affect plant growth and development (Schmitz, Grotewold & Stam, 2022). Phosphorylation signal transduction and phosphorus starvation response are affected by light, sugar, plant hormones (auxin, ethylene, cytokinin and gibberellin), as well as oxygen (Karthikeyan et al., 2007; Lei et al., 2011; Klecker et al., 2014). For example, the expression of AtPHR1 is regulated by light and ethylene, and the response to phosphorus starvation is regulated by the promoter of AtPHR gene (Liu et al., 2017). In this study, 13 types of cis-acting elements were identified in the promoter of PHR genes. A large number of light response elements and hormone elements showed that the expression and regulation of PHR genes were affected by light and hormone. MiRNAs regulate plant response to low phosphorus by down regulating gene transcription (Zeng et al., 2014). MiR399 and miR827 are involved in the response of plants to low phosphorus stress (Pant et al., 2008; Lin et al., 2010). In this study, 12 cotton miRNAs such as miR396, miR482 and miR827 have the potential to regulate GhPHR genes, which may play a role in phosphorus absorption and transport in cotton.

Most PHR genes in maize, rice and sorghum are continuously expressed in all tissues, indicating that they may play an important role in regulating phosphorus uptake and transport (Lin et al., 2013; Xu et al., 2018). This study analyzed the tissue expression of cotton PHR family genes in roots, stem, leaves and so on, and found that there was tissue-specific expression of cotton PHR family genes, which was similar to that of other crops (Lin et al., 2013). In tissue expression analysis, it was found that the expression of GhPHRs in root was high, and there were gibberellin and auxin response elements related to stress resistance in the cis-acting elements of promoter. In addition, the expression of GhPHRs exceeded the expression level before stress after low phosphorus stress, so it is speculated that GhPHRs may be related to the remodeling of root morphology under abiotic stress.

In conclusion, 128 PHR genes were identified in cotton, 41 of which were in G. hirsutum. There GhPHR genes had great differences in the number of amino acids and isoelectric point characteristics. In addition, the promoter region of GhPHRs has different cis-acting elements related to light response, and biotic and abiotic stresses. Further, expression analysis of the genes of showed that GhPHR11 and GhPHR18 were significantly highly expressed in root under low phosphorus stress. This study has provided a foundation for subsequent functional study of PHR genes and the breeding of new cotton varieties.

Supplemental Information

Supplemental Information 1 PHR genes in the four cotton species.

Click here for additional data file.

Supplemental Information 2 Raw data for qPCR.

Click here for additional data file.

Additional Information and Declarations

Competing Interests

Author Contributions

Data Availability

The authors declare that they have no competing interests.

Yan Zhao performed the experiments, analyzed the data, prepared figures and/or tables, and approved the final draft.

Peiyu Li performed the experiments, analyzed the data, prepared figures and/or tables, and approved the final draft.

Huarui Wang analyzed the data, prepared figures and/or tables, and approved the final draft.

Jiping Feng performed the experiments, prepared figures and/or tables, and approved the final draft.

Yuxin Li performed the experiments, prepared figures and/or tables, and approved the final draft.

Shanshan Wang performed the experiments, prepared figures and/or tables, and approved the final draft.

Yuanjie Li performed the experiments, prepared figures and/or tables, and approved the final draft.

Yanyan Guo performed the experiments, prepared figures and/or tables, and approved the final draft.

Lin Li performed the experiments, prepared figures and/or tables, and approved the final draft.

Yao Su conceived and designed the experiments, authored or reviewed drafts of the article, and approved the final draft.

Zhengwen Sun conceived and designed the experiments, authored or reviewed drafts of the article, and approved the final draft.

The following information was supplied regarding data availability:

The raw measurements are available in the Supplemental Files.

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
