# Peer review of "Genome-wide investigation and expression pattern of PHR family genes in cotton under low phosphorus stress"

_PeerJ, doi:10.7717/peerj.14584_

## Round 0.1 · original submission · Major Revisions

The article can be accepted for publication after major revisions.

·

Basic reporting

Thorough revision, especially of the INTRODUCTION, is recommended. Please see my comments and suggestions in the manuscript attached.

Experimental design

The research question is clear and direct, with a focus on exploring the molecular basis of cotton adaptation to phosphorus deficiency. I am satisfied with the research design, which I consider adequate to address the experimental question. The methods are described in sufficient detail and the rigour of their application is adequate.

Validity of the findings

The results are fairly novel, especially with regard to the key agronomic issue of managing cotton in phosphorus-deficient media; data are quite robust and relevant to the objectives. However, the discussion should more succinctly show how the present results compare with other [recent] similar studies, rather than reviewing the literature as it stands now.

Additional comments

See manuscript attached.

Reviewer 2 ·

Basic reporting

In this work Yan Zhao et. al. analyze PHR family genes in four species from Gossypium genus, additionally they evaluate this gene family gene expression in G. hirsutum under low phosphorus stress. The aim and scope of the article is very interesting and relevant. The manuscript clear and well writing.


The authors provide enough references and field background; however, the authors could take into account the next references:

Sega, P., & Pacak, A. (2019). Plant PHR transcription factors: put on a map. Genes, 10(12), 1018.

Lei, K., Cheng, J. Q., An, Y., Li, X. S., & An, G. (2022). Organ specific transcriptome analysis of upland cotton (Gossypium hirsutum) in response to low phosphorus stress during early stage of growth. Soil Science and Plant Nutrition, 1-10.

The structure of the article has a standard format. The figures are relevant and help to understand the paper.

Experimental design

Bioinformatics and experimental findings are original and within Aims and Scope of the journal. The identified knowledge gap (knowledge about omics information, evolution and expression patterns of PHR gene families in Gossypium genus) is well filled, following high technical standard. However, I recommend to perform a second phylogenetic reconstruction using a more robust method (Bayesian or ML) and select an adequate distance estimation model. Additionally, it could be considered to do the phylogenetic analysis using only the informative zones since members of a protein family of different size and with different conserved motifs are being used in the alignment.

Validity of the findings

The majority of the results and findings are well supported. However, there are some issues that must be clarified.

• In figure 1 bootstrap values are not showed to validate the five groups mentioned. Additionally, it is desirable to mark each group to a better understanding of the topic.
• In figure 2, the five groups mentioned are not distinguished, and the motif name are not showed.
• It was showed in the conserved domain analysis that domain content is variable, so the phylogenetic analysis could be repeated using a sequence alignment hand-made curated.
• In figure 4, experiments conditions must be described in methods.
• PHR gene family content, Phylogenetic analysis, Analysis of cis-acting elements and Prediction of miRNA-targets were poorly discussed; those sections must be re-written.
• Accession numbers from PHR transcription factor family of Arabidopsis thaliana proteins that were used as the reference sequences should be added.
• Accession numbers from genomes used should be added.
• In all figures legends would were poorly described and must be re-written.
• In all figures it is desirable to use Gene name to a better understanding of the topics.

Additional comments

Minor issues:

• There are systematic errors after a paper are be cited (example line 37, 50, 53, and so on).
• Line 124, Subtopic are missing.
• In table 1, D subgenome are missing.
• In table 1, scientific name should be in italics characters.

Reviewer 3 ·

Basic reporting

Phosphorus starvation response (PHR) protein is an important transcription factor in
phosphorus regulatory network, which plays an important role in regulating the utilization of phosphorus. So far, the PHR genes were systematically investigated in cotton, which was very valuable.

Experimental design

The manuscript has reasonable experimental design. However, obviously, the manuscript was written rather carelessly and illogically.

Validity of the findings

(1) The information presented in the following places in your manuscript is illogically and should be rephrased, including lines 149; line 167 to line 172; lines 183.
(2) Species and gene names should be italicized, in line 177 “13 Arabidopsis PHR amino acid sequences” should be revised to “13 Arabidopsis PHR amino acid sequences”; in line 222 “the regulatory miRNAs of PHR” should be revised to “the regulatory miRNAs of PHR”. Such cases should be checked and corrected in full.
(2) The image resolution is too low. e.g., Figures 2 and 5.
(3) The drawings in this manuscript do not agree with the text. In lines 253 to 266, this part only describes the expression pattern of PHR gene in different tissues. However, the expression pattern of PHR gene in different stresses is also shown in Figure 4. Please keep it consistent.
(4) The writing of the manuscript is not careful. For example, in line 2 “werehighly expressed in all tissues” should be revised to “were highly expressed in all tissues”.
(5) In References, Need to double-check the formatting as there are some inconsistencies. For instance, lines 343-344, the This document lacks volume, issue, page number, etc.

Additional comments

No

Reviewer 4 ·

Basic reporting

no comments

Experimental design

no comments

Validity of the findings

no comments

Additional comments

1. The genes’ name in the manuscript, figures and tables are not uniform, please modify it.
2. In figure 1, the authors wrote “The specific color indicated different families”, but the figure 1 does not show the five subgroups.
3. Line 479, Line 482, the “color scale” should be “colorful scale”. 4. The authors should check the grammar erros.
5. The authors should point the reasons why choose the six GhPHRs to do qPCR.
6. I think the qPCR might be qRT-PCR, please check.

---

## Round 0.2 · Minor Revisions

According to the comments of the reviewer, the manuscript can be considered to be accepted for publication after minor modifications.

Reviewer 2 ·

Basic reporting

This is the second revision of this paper.

The aim and scope of the article is very interesting and relevant. The manuscript clear and well writing. The authors provide enough references and field background.

The structure of the article has a standard format. The figures are relevant and help to understand the paper.

Experimental design

Bioinformatics and experimental findings are original and within Aims and Scope of the journal.

In the response letter, the authors mention a second phylogenetic reconstruction, however no evidence of this phylogenetic tree has been added. In the same way, I recommend publishing the ML tree with the bootstrap values over 50 (0.5) and add the methodology to the respective section.

Validity of the findings

The majority of the results and findings are well supported. However, there are some issues that must be clarified.

• What is phylogenetic distribution telling us about protein function?
• It is a motif content relation to protein function?
• It is a cis-element content relation to gene expression?

Minor issues:

• In figure 1 legend. “The specific color indicated different families”; what did it means? It appears to be a species distribution where Arabidopsis and G. raimondii are indicated in the same color.
• In figure 2. Add known-motif name (Myb_DNA-binding, Myb_CC_LHEQLE) to the figure to clarify.
• In figure 3, the five groups mentioned are not distinguished.
• In figure 1, Remove bootstrap values below 0.5. For a better visualization, only show two decimal values and move the values that overlap the lines.
• In figure 2. Add panel A and B and explain separately in legend.
• In figure 3 legend. Explain figure.
• In figure 5. Add panel A and B and explain separately in legend.
• In figure 6. Add panel A … F and explain separately in legend.

---

## Round 0.3 · accepted · Accept

According to the comments of the reviewer, the manuscript can be considered to be accepted for publication.